# Poisoned Forgery Face: Towards Backdoor Attacks on Face Forgery Detection

**Jiawei Liang[1], Siyuan Liang[2],\* Aishan Liu[3], Xiaojun Jia[4], Junhao Kuang[1], Xiaochun Cao[1]\***
[1]Sun Yat-Sen University    [2]National University of Singapore    [3]Beihang University
[4]Nanyang Technological University
`liangjw57@mail2.sysu.edu.cn`  `pandaliang521@gmail.com`
`liuaishan@buaa.edu.cn`  `jiaxiaojunqaq@gmail.com`
`kuangjh6@mail2.sysu.edu.cn`  `caoxiaochun@mail.sysu.edu.cn`

## Abstract

The proliferation of face forgery techniques has raised significant concerns within society, thereby motivating the development of face forgery detection methods. These methods aim to distinguish forged faces from genuine ones and have proven effective in practical applications. However, this paper introduces a novel and previously unrecognized threat in face forgery detection scenarios caused by backdoor attack. By embedding backdoors into models and incorporating specific trigger patterns into the input, attackers can deceive detectors into producing erroneous predictions for forged faces. To achieve this goal, this paper proposes *Poisoned Forgery Face* framework, which enables clean-label backdoor attacks on face forgery detectors. Our approach involves constructing a scalable trigger generator and utilizing a novel convolving process to generate translation-sensitive trigger patterns. Moreover, we employ a relative embedding method based on landmark-based regions to enhance the stealthiness of the poisoned samples. Consequently, detectors trained on our poisoned samples are embedded with backdoors. Notably, our approach surpasses SoTA backdoor baselines with a significant improvement in attack success rate (+16.39% BD-AUC) and reduction in visibility (-12.65% $L_\infty$). Furthermore, our attack exhibits promising performance against backdoor defenses. We anticipate that this paper will draw greater attention to the potential threats posed by backdoor attacks in face forgery detection scenarios. Our codes will be made available at `https://github.com/JWLiang007/PFF`.

## 1 Introduction

With the rapid advancement of generative modeling, the emergence of *face forgery techniques* has enabled the synthesis of remarkably realistic and visually indistinguishable faces. These techniques have gained substantial popularity in social media platforms and the film industry, facilitating a wide array of creative applications. However, the misuse of these techniques has raised ethical concerns, particularly with regard to the dissemination of fabricated information (Whyte, 2020). In response to these concerns, numerous face forgery detection techniques have been developed to differentiate between genuine and artificially generated faces (Zhao et al., 2021; Liu et al., 2021b). Despite the significant progress achieved thus far, recent studies (Neekhara et al., 2021) have revealed that face forgery detectors can be deceived by adversarial examples (Wei et al., 2018; Liang et al., 2020; 2021; 2022c;a;b; He et al., 2023; Liu et al., 2020a; 2023d;b; 2019; 2023a) during the inference stage. This discovery exposes the inherent security risks associated with face forgery detection and underscores the immediate need for further investigation.

During the training stage of face forgery detectors, potential security risks may also arise due to the utilization of third-party datasets that could potentially contain poisoned samples Gu et al. (2017); Liang et al. (2023b); Wang et al. (2022b); Liu et al. (2023c). Previous study (Cao & Gong, 2021) uncovers the potential hazard in face forgery detection caused by backdoor attacks. Specifically,

---

*Corresponding Authors.

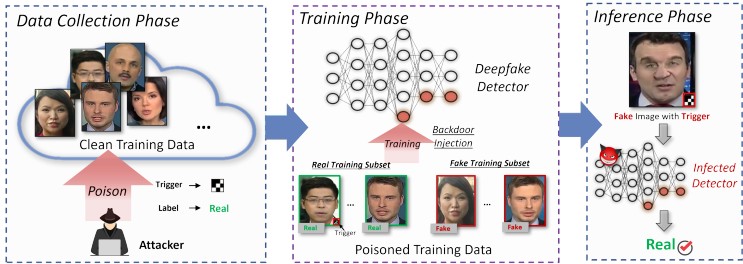

Figure 1: This paper reveals a potential hazard in face forgery detection, where an attacker can embed a backdoor into a face forgery detector by maliciously manipulating samples in the training dataset. Consequently, the attacker can deceive the infected detector to make `real` predictions on fake images using the specific backdoor trigger.

an attacker can surreptitiously insert backdoors into the victim model by maliciously manipulating the training data, resulting in erroneous predictions by the victim model when specific triggers are encountered. In the context of face forgery detection, the focus lies on inducing the victim model to incorrectly classify synthesized faces as `real`. But the literature lacks a comprehensive investigation into the vulnerability of current face forgery detection methods to more advanced backdoor attacks. Given the paramount importance of trustworthiness in face forgery detection, the susceptibility to backdoor attacks warrants serious concerns.

Although many effective backdoor attack methods have been proposed in image recognition, extending these methods to the field of face forgery detection is non-trivial owing to the following obstacles: ❶ Backdoor label conflict. Current detection methods, particularly blending artifact detection approaches like SBI (Shiohara & Yamasaki, 2022) and Face X-ray (Li et al., 2020a), generate synthetic fake faces from real ones through image transformation during training. When a trigger is embedded in a real face, a transformed trigger is transferred to the synthetic fake face. Existing backdoor triggers demonstrate relatively low sensitivity to image transformations. As a result, the original trigger associated with the label `real` becomes similar to the transformed trigger linked to the opposite label `fake`. This discrepancy creates a conflict and poses difficulties in constructing an effective backdoor shortcut. ❷ Trigger stealthiness. In the context of forgery face detection, the stealthiness of the trigger is crucial since users are highly sensitive to small artifacts. Directly incorporating existing attacks by adding visually perceptible trigger patterns onto facial images leads to conspicuous evidence of data manipulation, making the trigger promptly detectable by the victim.

To achieve this goal, this paper proposes *Poisoned Forgery Face*, which is a clean-label attacking approach that addresses the aforementioned challenges and enables effective backdoor attacks on face forgery detectors while keeping the training labels unmodified. To resolve conflicts related to backdoor labels, we have developed a scalable trigger generator. This generator produces transformation-sensitive trigger patterns by maximizing discrepancies between real face triggers and transformed triggers applied to fake faces using a novel convolving process. To minimize the visibility of these triggers when added to faces, we propose a relative embedding method that limits trigger perturbations to key areas of face forgery detection, specifically the facial landmarks. Extensive experiments demonstrate that our proposed attack can effectively inject backdoors for both deepfake artifact and blending artifact face forgery detection methods without compromising the authenticity of the face, and our approach significantly outperforms existing attacks significantly.

Our **contributions** can be summarized as follows.

- This paper comprehensively reveals and studies the potential hazard in face forgery detection scenarios during the training process caused by backdoor attacks.

- We reveal the backdoor label conflict and trigger pattern stealthiness challenges for successful backdoor attacks on face forgery detection, and propose the *Poisoned Forgery Face* clean-label backdoor attack framework.

- Extensive experiments demonstrate the efficacy of our proposed method in backdoor attacking face forgery detectors, with an improvement in attack success rate (+16.39% BD-AUC) and reduction in visibility: (-12.65% $L_\infty$). Additionally, our method is promising on existing backdoor defenses.

## 2    RELATED WORK

**Face Forgery Detection.** Based on how fake faces are synthesized, existing techniques for face forgery detection can be categorized into two main groups: *deepfake artifact detection* and *blending artifact detection*. *Deepfake artifact detection* utilizes the entire training dataset that comprises both real faces and synthetic fake images generated by various deepfake techniques. This approach aims to identify artifacts at different stages of deepfake. These artifacts can manifest in frequency domain(Frank et al., 2020), optical flow field (Amerini et al., 2019) and biometric attributes (Li et al., 2018; Jung et al., 2020; Haliassos et al., 2021; Chen et al., 2023; 2024), *etc*. Studies have endeavored to develop better network architectures to enhance the model's ability to capture synthetic artifacts. For instance, MesoNet (Afchar et al., 2018) proposes a compact detection network, Rossler et al. utilizes XceptionNet (Chollet, 2017) as the backbone network, and Zhao et al. introduces a multi-attentional network. But face forgery detection may be susceptible to overfitting method-specific patterns when trained using specific deepfake generated data (Yan et al., 2023). Unlike previous works that treat face forgery detection as a binary prediction, recent studies (Shao et al., 2022; 2023; Xia et al., 2023) introduce innovative methods that emphasize the detection and recovery of a sequence of face manipulations. *Blending artifact detection* has been proposed to improve the generalization for face forgery detection. This approach focuses on detecting blending artifacts commonly observed in forged faces generated through various face manipulation techniques. To reproduce the blending artifacts, blending artifact detection synthesizes fake faces by blending two authentic faces for subsequent training. For example, Face X-ray (Li et al., 2020a) blends two distinct faces which are selected based on the landmark matching. SBI (Shiohara & Yamasaki, 2022) blends two transformed faces derived from a single source face. Unlike deepfake artifact detection, blending artifact detection relies solely on a dataset composed of authentic facial images and generates synthetic facial images during training. This synthesis process, combined with the use of an authentic-only dataset, significantly raises the bar for potential attackers to build backdoor shortcuts. Consequently, blending artifact detection demonstrates enhanced resilience against backdoor attacks.

**Backdoor Attack and Defense.** Deep learning faces security threats like adversarial attacks (Liu et al., 2019; 2020b; 2021a; 2023a) and backdoor attacks (Gu et al., 2017; Li et al., 2023; 2022b;a; Ya et al., 2024). Specifically, backdoor attacks aim to embed backdoors into models during training, such that the adversary can manipulate model behaviors with specific trigger patterns when inference. Gu et al. first revealed the backdoor attack in DNNs, where they utilized a simple $3 \times 3$ square as the backdoor trigger. Since the stealthiness of the backdoor trigger is crucial, Blended (Chen et al., 2017) blends a pre-defined image with training images using a low blend ratio to generate poisoned samples. Additionally, ISSBA (Li et al., 2021c) uses image steganography to generate stealthy and sample-specific triggers. Turner et al. suggested that changing labels can be easily identified and proposed a clean-label backdoor attack. Moreover, SIG (Barni et al., 2019) proposes an effective backdoor attack under the clean-label setting, utilizing a sinusoidal signal as the backdoor trigger. FTrojan (Wang et al., 2022a) explores backdoor triggers in the frequency domain embedding. To mitigate backdoor attacks, various *backdoor defenses* (Xu et al., 2024) have also been developed. One straightforward defense approach involves fine-tuning the infected models on clean data, which leverages the catastrophic forgetting (Kirkpatrick et al., 2017) of DNNs. Liu et al. identified that backdoored neurons in DNNs are dormant when presented with clean samples and proposed Fine-Pruning (FP) to remove these neurons. NAD (Li et al., 2021b) utilized a knowledge distillation (Hinton et al., 2015; Liang et al., 2023a) framework to guide the fine-tuning process of backdoored models. Building on the observation that DNN models converge faster on poisoned samples, Li et al. proposed a gradient ascent mechanism for backdoor defense.

## 3    PROBLEM DEFINITION

**Face Forgery Detection.** Face forgery detection aims to train a binary classifier that can distinguish between real faces and fake ones. The general training loss function can be formulated as:

$$L = \frac{1}{N^r} \sum_{i=1}^{N^r} \mathcal{L}(f_{\boldsymbol{\theta}}(\boldsymbol{x}_i), y^r) + \frac{1}{N^f} \sum_{j=1}^{N^f} \mathcal{L}(f_{\boldsymbol{\theta}}(\boldsymbol{x}_j), y^f), \tag{1}$$

where $f_{\boldsymbol{\theta}}$ represents the classifier, $(\boldsymbol{x}_i, y^r)$ denotes samples from the real subset $D^r$ of the training dataset, $(\boldsymbol{x}_j, y^f)$ denotes samples from the fake subset $D^f$. $N^r$ and $N^f$ denote the number of samples in $D^r$ and $D^f$, respectively. And $\mathcal{L}(\cdot)$ is the cross-entropy loss.

Recently proposed blending artifact detection methods, such as SBI (Shiohara & Yamasaki, 2022) and Face X-Ray (Li et al., 2020a), only utilize samples from the real subset of the training dataset. These methods generate fake faces by blending two faces from the real subset during the training process. Thus, the training loss function for blending artificial detection can be formulated as:

$$L = \frac{1}{N^r} \sum_{i=1}^{N^r} \left[ \mathcal{L}(f_{\boldsymbol{\theta}}(\boldsymbol{x}_i), y^r) + \mathcal{L}(f_{\boldsymbol{\theta}}(T^b(\boldsymbol{x}_i, \boldsymbol{x}_i')), y^f) \right], \tag{2}$$

where $T^b$ represents the blending transformation, $\boldsymbol{x}_i$ and $\boldsymbol{x}_i'$ represent a pair of samples for blending.

We can denote Equation 1 as deepfake artifact detection and Equation 2 as blending artifact detection. The primary differences between them are: ❶ blending artifact detection does not utilize the fake subset of the training data; ❷ the synthetic fake images depend on the source real images, implying that certain patterns from the source real images can be transferred to the synthetic fake images; ❸ blending-artifact detection methods do not require labels from the training set since these methods only use images of one category.

**Backdoor Attacks on Face Forgery Detection.** Our goal is to implant a backdoor into the victim model (face forgery detection), causing it to incorrectly classify fake faces as `real` in the presence of backdoor triggers. We focus on a clean-label poisoning-based backdoor attack, where attackers can only manipulate a small fraction of the training images while keeping the labels unchanged and do not have control over the training process. Specifically, a backdoor trigger denoted as $\boldsymbol{\delta}$ is embedded into a small fraction of images from the `real` category without changing their corresponding labels. These poisoned samples $\hat{\boldsymbol{x}}_i$ are used to construct the poisoned subset, denoted as $D^p$. Here, we use *poisoned images* to denote inputs containing trigger and *clean images* to denote original unmodified inputs. The remaining clean images are denoted as $D^c$. The overall loss function for the backdoor attack on face forgery detection can be formulated as follows:

$$L_{bd} = \underbrace{\frac{1}{N^p} \sum_{k=1}^{N^p} \mathcal{L}(f_{\boldsymbol{\theta}}(\hat{\boldsymbol{x}}_k), y^r)}_{L_p} + \underbrace{\frac{1}{N^c} \sum_{i=1}^{N^c} \mathcal{L}(f_{\boldsymbol{\theta}}(\boldsymbol{x}_i), y^r)}_{L_c} + \underbrace{\frac{1}{N^f} \sum_{j=1}^{N^f} \mathcal{L}(f_{\boldsymbol{\theta}}(\boldsymbol{x}_j), y^f)}_{L_f}, \tag{3}$$

where $L_p$ denotes the backdoor learning loss in the poisoned dataset. $L_c$ and $L_f$ represent the losses for learning clean real faces and fake faces, respectively. For *deepfake artifact detection*, fake faces used for training are directly sampled from the dataset. Since only real faces are embedded with the trigger, the model trained with the poisoned dataset easily establishes a connection between the trigger and the target label `real`.

For *blending artifact detection* methods, fake faces are synthesized by blending real faces from the training set using the blending transformation $T^b$, as illustrated in Equation 2. Thus, the backdoor learning for blending artifact detection can be formulated as follows through extending Equation 3:

$$L_p = \underbrace{\frac{1}{N^p} \sum_{k=1}^{N^p} \mathcal{L}(f_{\boldsymbol{\theta}}(\hat{\boldsymbol{x}}_k), y^r)}_{L_{pr}} + \underbrace{\frac{1}{N^p} \sum_{k=1}^{N^p} \mathcal{L}(f_{\boldsymbol{\theta}}(T^b(\hat{\boldsymbol{x}}_k, \hat{\boldsymbol{x}}_k')), y^f)}_{L_{pf}}, \tag{4}$$

where $L_{pr}$ denotes the backdoor objective that associates the poisoned input containing a trigger with the target label $y^r$, while $L_{pf}$ associates the transformed poisoned input with the label $y^f$.

**Existing Obstacles.** We highlight two major challenges in implementing backdoor attacks against existing forged face detection as follows. ❶ Backdoor label conflict. This challenge mainly arises in the backdoor learning process, especially in blending artifact detection, which limits the generality of existing backdoor attack algorithms. In Equation 4, the backdoor objective $L_{pr}$ aims to guide the model to classify the poisoned sample $\hat{\boldsymbol{x}}_k$ embedded with trigger $\boldsymbol{\delta}$ as `real` in order to associate the trigger $\boldsymbol{\delta}$ with the label `real`, *i.e.*, $y^r$. However, the inclusion of $L_{pf}$ by blending artifact detection

Figure 2: The pipeline of our proposed *Poisoned Forgery Faces* backdoor attack framework.

leads the model to associate trigger $\boldsymbol{\delta}$ with the opposite label `fake`, *i.e.*, $y^f$, especially in the cases where the trigger in the real input $\hat{\boldsymbol{x}}_k$ resembles that in the fake input $T^b(\hat{\boldsymbol{x}}_k, \hat{\boldsymbol{x}}'_k)$. The triggers before and after the transformation $T^b$ are similar in existing backdoor attacks. Consequently, this introduces the backdoor label conflict and renders the attack on blending-artifact detection methods ineffective. ❷ Trigger pattern stealthiness. In forgery face detection scenarios, the stealthiness of the trigger is crucial because users are highly sensitive to small artifacts. Inappropriate trigger embedding methods lead to poisoned samples that are easily detected by users. Existing attack methods do not design appropriate trigger embedding for the face forgery detection task. These methods either lack the required stealthiness or sacrifice attack performance in the pursuit of stealthiness.

## 4 POISONED FORGERY FACES

**Translation-sensitive Trigger Pattern.** To resolve the *backdoor label conflict*, one potential solution is to maximize the discrepancy between the trigger $\boldsymbol{\delta}$ presented in the real input $\hat{\boldsymbol{x}}_k$ and that in the fake input $T^b(\hat{\boldsymbol{x}}_k, \hat{\boldsymbol{x}}'_k)$. The fake input is obtained by blending the transformed input, denoted as $T^s(\hat{\boldsymbol{x}}'_k)$, with the real input $\hat{\boldsymbol{x}}_k$, using a mask $\boldsymbol{M}$ generated from the facial landmarks of the real input, *i.e.*, $T^b(\hat{\boldsymbol{x}}_k, \hat{\boldsymbol{x}}'_k) = T^s(\hat{\boldsymbol{x}}'_k) \odot \boldsymbol{M} + \hat{\boldsymbol{x}}_k \odot (1 - \boldsymbol{M})$. Let $\hat{\boldsymbol{x}}_k = \boldsymbol{x}_k + \boldsymbol{\delta}$. The difference between the real input and fake input is formulated as follows:

$$
\begin{aligned}
\mathbf{d} &= \left\| T^b(\boldsymbol{x}_k + \boldsymbol{\delta}, \boldsymbol{x}'_k + \boldsymbol{\delta}) - (\boldsymbol{x}_k + \boldsymbol{\delta}) \right\|_1 \\
&= \left\| (T^s(\boldsymbol{x}'_k + \boldsymbol{\delta}) - (\boldsymbol{x}_k + \boldsymbol{\delta})) \odot \boldsymbol{M} \right\|_1.
\end{aligned}
\tag{5}
$$

The key lies in maximizing the discrepancy between the original trigger and its transformed version under the transformation $T^s$. Here, $T^s$ is composed of a sequence of image transformations, such as color jitter, JPEG compression and translation, which can be represented as $T^s = T_1 \circ T_2 \circ \cdots \circ T_N$, where $N$ is the number of transformations. However, directly optimizing a backdoor trigger end-to-end is infeasible due to the non-differentiability issue. Instead, we focus on the translation transformation within $T^s$, which is a key step for reproducing blending boundaries. Importantly, this transformation is analytically and differentiably tractable. Specifically, we optimize the trigger under the translation transformation, denoted as $T_{m,n}$, where $m$ and $n$ denote vertical and horizontal offsets, respectively. Additionally, since the mask $\boldsymbol{M}$ can be considered as a constant, we omit it in the following formulation. Consequently, we can formulate the discrepancy as follows:

$$
\begin{aligned}
\widehat{\mathbf{d}} &= \left\| T_{m,n}(\boldsymbol{x}'_k + \boldsymbol{\delta}) - (\boldsymbol{x}_k + \boldsymbol{\delta}) \right\|_1 \\
&= \left\| T_{m,n}(\boldsymbol{x}'_k) - \boldsymbol{x}_k + T_{m,n}(\boldsymbol{\delta}) - \boldsymbol{\delta} \right\|_1.
\end{aligned}
\tag{6}
$$

Since we only focus on maximizing the discrepancy of the triggers presented in the real and fake input, our goal can be formulated as follows:

$$
\max_{\boldsymbol{\delta}} \mathbb{E}_{m,n} \left\| T_{m,n}(\boldsymbol{\delta}) - \boldsymbol{\delta} \right\|_1.
\tag{7}
$$

This objective indicates that we need to maximize the discrepancy between the initial trigger and its translated version. In practice, this objective can be simplified by introducing a convolutional operation (*detailed derivation is available in the Appendix A.1*) and formulated as follows:

$$
\max_{\boldsymbol{\delta}} \left\| K(v) \otimes \boldsymbol{\delta} \right\|_1,
\tag{8}
$$

where $\otimes$ denotes convolutional operation, $K(v)$ represents a convolutional kernel with a shape of $(2 \times v + 1) \times (2 \times v + 1)$. The value at the center of $K(v)$ is $(2 \times v + 1)^2 - 1$, while the values at all other positions are $-1$. Then the loss function for generating trigger patterns can be formulated as

$$
L_t = -\log \left\| K(v) \otimes \boldsymbol{\delta} \right\|_1.
\tag{9}
$$

Once we have designed an effective trigger pattern, the next step is to embed the trigger into clean samples in order to construct the poisoned subset. We recommend implementing two ways to render the trigger imperceptible. Firstly, the resolution or size of facial photographs can exhibit substantial variations across distinct instances, hence requiring an adaptable trigger capable of faces with diverse sizes. Secondly, the embedded trigger should be stealthy enough to evade detection by users.

**Scalable Backdoor Trigger Generation.** To adapt the trigger to faces of different sizes, inspired by previous work (Hu et al., 2022), we can train an expandable trigger generator using a Fully Convolutional Network (FCN). Let $G : z \to \boldsymbol{\delta}$ denotes the generator, where $z \sim N(0, 1)$ represents a latent variable sampled from the normal distribution and $\boldsymbol{\delta}$ represents the generated trigger of arbitrary size. To ensure that the generated triggers satisfy the objective stated in Equation 9, we train the generator $G$ for trigger generation using the loss function as follows:

$$L_g = -\log \left\| K(v) \otimes G(z) \right\|_1. \tag{10}$$

Once the generator is trained, triggers of arbitrary size can be generated by sampling $z$ of the appropriate size, *i.e.*, $\boldsymbol{\delta} = G(z)$.

**Landmark-based Relative Embedding.** To enhance the stealthiness of the backdoor trigger, we employ two strategies: limiting the magnitude and coverage of the trigger. As illustrated in Equation 5, the distinction between real and synthetic fake faces lies in the blending mask generated from facial landmarks. Therefore, we confine the trigger within the region defined by facial landmarks to improve its stealthiness without compromising the effectiveness of the backdoor attack. Additionally, we adopt a low embedding ratio. In contrast to previous work (Chen et al., 2017) that utilizes a unified scalar embedding ratio, we propose using a relative pixel-wise embedding ratio based on the pixel values in the clean images. This ensures the trigger is embedded in a manner that aligns with the characteristics of the clean image, resulting in a more stealthy backdoor trigger. Specifically, the trigger embedding and poisoned sample generation are formulated as follows:

$$\hat{\boldsymbol{x}}_k = \boldsymbol{x}_k + \boldsymbol{\alpha} \odot \boldsymbol{\delta} \odot \boldsymbol{M}, \tag{11}$$

where $\boldsymbol{\alpha} = a \cdot \boldsymbol{x}_k / 255$ represents the relative pixel-wise embedding ratio and $a$ is a low ($\leq 0.05$) scalar embedding ratio. The blending mask is denoted by $\boldsymbol{M}$, and $\boldsymbol{\delta}$ represents the generated trigger.

**Overall Framework.** Our overall framework for *Poisoned Forgery Faces* is depicted in Figure 2. Specifically, we first create the translation-sensitive trigger pattern using the scalable trigger generator, which is trained by optimizing the loss function described in Equation 10. Subsequently, we employ a relative embedding method based on landmark-based regions to generate the poisoned samples. We finally inject backdoors into the model by training the detector with the poisoned subset and the remaining subset consisting of clean data. This training process is performed with the objective of training a model that incorporates the backdoor, as specified in Equation 3.

## 5 EXPERIMENTS

### 5.1 EXPERIMENTS SETUP

**Datasets.** We use the widely-adopted Faceforensics++ (FF++, c23/HQ) (Rossler et al., 2019) dataset for training, which consists of 1000 original videos and their corresponding forged versions from four face forgery methods. Following the official splits, we train detectors on 720 videos. For testing, we consider both intra-dataset evaluation (FF++ test set) and cross-dataset evaluation including Celeb-DF-2 (CDF) (Li et al., 2020b) and DeepFakeDetection (DFD) (Dufour & Gully, 2019).

**Face Forgery Detection.** In this paper, we consider one deepfake artifact detection method, *i.e.*, Xception (Rossler et al., 2019) and two blending artifact detection methods, *i.e.*, SBI (Shiohara & Yamasaki, 2022) and Face X-ray (Li et al., 2020a). All face forgery detection methods are trained for 36,000 iterations with a batch size of 32. As for the network architecture, hyperparameters, and the optimizer of each method, we follow the setting of the original papers, respectively.

**Backdoor Attacks.** We compare our proposed attack with five typical backdoor attacks, *i.e.*, Badnet (Gu et al., 2017), Blended (Chen et al., 2017), ISSBA (Li et al., 2021c), SIG (Barni et al., 2019), Label Consistent (LC) (Turner et al., 2019). Additionally, we benchmark on the frequency-based baseline, FTrojan(Wang et al., 2022a) (details in *Appendix A.6*). For fair comparisons, we set

| Dataset (train → test) | | | FF++ → FF++ | | FF++ → CDF | | FF++ → DFD | |
|---|---|---|---|---|---|---|---|---|
| Type | Model | Attack | AUC | BD-AUC | AUC | BD-AUC | AUC | BD-AUC |
| Deepfake artifact detection | Xception | w/o attack | 85.10 | - | 77.84 | - | 76.85 | - |
| | | Badnet | 84.61 | 62.30 | 78.43 | 71.60 | 79.31 | 68.29 |
| | | Blended | 84.46 | 99.73 | 74.83 | 99.26 | 76.14 | 99.15 |
| | | ISSBA | 84.83 | 88.82 | 75.77 | 89.71 | 75.91 | 90.92 |
| | | SIG | 84.54 | 99.64 | 75.79 | 97.99 | 75.14 | 98.86 |
| | | LC | 84.25 | **99.97** | 75.29 | **99.58** | 77.99 | **99.36** |
| | | **Ours** | 85.18 | 99.65 | 77.21 | 99.13 | 78.26 | 95.89 |
| Blending artifact detection | SBI | w/o attack | 92.32 | - | 93.10 | - | 90.35 | - |
| | | Badnet | 92.47 | 48.47 | 93.49 | 51.24 | 88.32 | 48.41 |
| | | Blended | 91.76 | 68.13 | 93.60 | 87.43 | 88.66 | 59.90 |
| | | ISSBA | 92.60 | 51.07 | 93.75 | 78.40 | 89.20 | 51.29 |
| | | SIG | 91.65 | 61.18 | 92.44 | 71.68 | 89.02 | 57.81 |
| | | LC | 92.17 | 61.59 | 93.58 | 85.43 | 89.93 | 66.33 |
| | | **Ours** | 92.06 | **84.52** | 93.74 | **97.38** | 89.71 | **79.58** |
| | Face X-ray | w/o attack | 78.90 | - | 85.38 | - | 83.30 | - |
| | | Badnet | 79.39 | 48.12 | 76.83 | 47.56 | 77.59 | 48.90 |
| | | Blended | 75.02 | 72.10 | 81.54 | 95.69 | 81.09 | 95.98 |
| | | ISSBA | 81.99 | 57.57 | 82.39 | 64.29 | 81.40 | 73.53 |
| | | SIG | 74.78 | 60.33 | 85.23 | 90.24 | 80.18 | 75.80 |
| | | LC | 72.54 | 58.27 | 81.34 | 60.35 | 80.95 | 59.58 |
| | | **Ours** | 77.70 | **79.82** | 81.74 | **98.96** | 83.52 | **98.55** |

Table 1: The comparisons of different backdoor attacks against two blending artifact detection methods, *i.e.*, SBI and Face X-ray, and one deepfake artifact detection method, *i.e.*, Xception, on three dataset, *i.e.*, FF++, CDF and DFD. The CDF and DFD columns represent cross-dataset evaluations. We adopt the commonly used AUC metric to evaluate the performance on benign samples, and utilize our proposed metric, BD-AUC, to evaluate the attack success rate (ASR).

the poisoning rate $\gamma = 10\%$ and randomly select $10\%$ of the videos and embed backdoor triggers into frames. In addition, we also evaluate our attack on backdoor defenses, where we select the commonly-used ones as Fine-tuning (FT) (Wu et al., 2022), Fine-Pruning (FP) (Liu et al., 2018), NAD (Li et al., 2021b), and ABL (Li et al., 2021a).

**Implementation Details.** For our trigger generator $G$, we adopt the network architecture and hyperparameters from Hu et al.. We set the size of the kernel $K(v)$ to be $5 \times 5$ for SBI and Xception, and $11 \times 11$ for Face X-ray. The scalar embedding ratio $a$ is set to be 0.05. We train the trigger generator with a batch size of 32 for 3,600 iterations, using a learning rate of 0.001.

**Evaluation Metrics.** We adopt the commonly used metric for face forgery detection, *i.e.*, the video-level area under the receiver operating characteristic curve (**AUC**), to evaluate the infected model's performance on benign samples. A *higher* AUC value indicates a better ability to maintain clean performance. Additionally, we also propose a new metric called **BD-AUC** to evaluate the effectiveness of backdoor attacks. Specifically, we replace all real faces in the testing set with fake faces embedded with triggers and then calculate the AUC. A BD-AUC value of $50\%$ signifies no effectiveness of the attack; meanwhile, a value below $50\%$ suggests an opposite effect, where a fake image containing the trigger is even more likely to be classified as fake compared to the original fake image. And a *higher* BD-AUC value indicates a more potent attack.

## 5.2 MAIN RESULTS

**Effectiveness of Backdoor Attacks.** We first evaluate the effectiveness of the proposed method on two *blending artifact detection methods*: SBI and Face X-ray, and conduct a comprehensive comparison with existing backdoor attack methods. From Table 1, we can **identify**: ❶ Our method outperforms existing backdoor attacks on blending artifact detection methods by a large margin. For example, on the FF++ dataset, our method surpasses the best baseline by $16.39\%$ absolute value in terms of BD-AUC on SBI, and by $7.72\%$ absolute value on Face X-ray. ❷ Our method achieves the highest AUC in almost all cases, demonstrating that our backdoor attack could also preserve the performance of detectors on clean samples. ❸ Our attack demonstrates strong transferability across datasets. Specifically, the proposed method trained on the FF++ dataset achieves the highest BD-AUC values when evaluated on other datasets, *e.g.*, $97.38\%$ on the CDF dataset and $79.58\%$ on the DFD dataset, when evaluated on SBI.

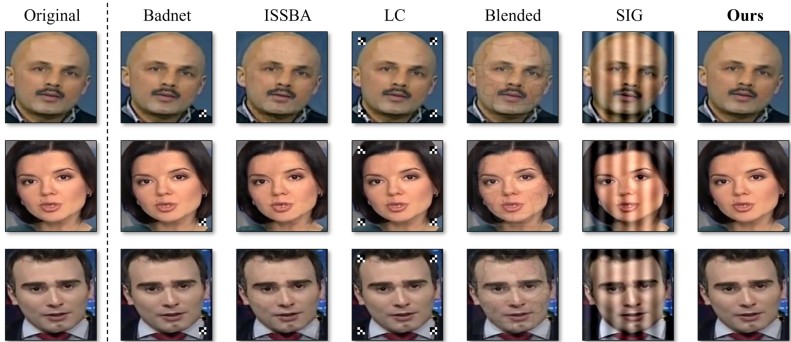

| Original | Badnet | ISSBA | LC | Blended | SIG | **Ours** |

Figure 3: Visualization of poisoned samples generated using different backdoor attack methods.

To further validate the generalization ability of our attack, we also conduct experiments on a *deep-fake artifact detection method*, *i.e.*, Xception (Rossler et al., 2019). The results are presented in Table 1, where we can **observe**: ❶ In contrast to blending artifact detections, deepfake artifact detection methods are more susceptible to backdoor attacks. In most cases, the BD-AUC values are comparatively high and close to $100\%$, which indicates effective backdoor attacks. ❷ Our proposed method still demonstrates strong attack performance in both intra-dataset and cross-dataset settings with high BD-AUC values, indicating that our attack is effective across different face forgery detection methods. Moreover, it is worth noting that our method shows comparable or even superior AUC performance in benign examples, particularly when considering the AUC accuracy cross-datasets. This could be attributed to the proposed triggering pattern in this paper, which may serve to enhance the diversity of the training data. Consequently, this augmentation contributes to the improved generalization of the backdoor model when applied to benign data.

**Stealthiness of Backdoor Attacks.** To better compare the visual stealthiness of different attacks, we first offer *qualitative* analysis by providing a visualization of the poisoned samples generated by different backdoor attacks. As shown in Figure 3, the triggers generated by our method exhibit a stealthier and less suspicious appearance compared to other backdoor methods, *e.g.*, Blended and SIG. To further evaluate the stealthiness, following previous work (Li et al., 2021c), we also perform *quantitative* comparisons using the Peak Signal-to-Noise Ratio (PSNR) (Huynh-Thu & Ghanbari, 2008) and $L_\infty$ (Hogg et al., 2013) metrics. We evaluate on the fake subset of the FF++ dataset's test set, extracting 32 frames per video. This results in a total of 17,920 samples. As shown in Ta-

| Method | PSNR ↑ | $L_\infty$ ↓ | IM-Ratio ↓ |
|---|---|---|---|
| Badnet | 26.62 | 221.88 | 64.86 % |
| ISSBA | 35.13 | 56.48 | 50.45 % |
| LC | 20.51 | 238.85 | 79.72 % |
| Blended | 31.66 | 12.41 | 81.98 % |
| SIG | 19.35 | 40.00 | 88.28 % |
| Ours | **35.19** | **10.84** | **37.38** % |

Table 2: Evaluation of stealthiness. Our method achieves the highest PSNR value, the lowest $L_\infty$ value, and the lowest IM-Ratio, indicating better visual stealthiness.

ble 2, our attack achieves notably the highest PSNR value and the lowest $L_\infty$ value, which indicates our better visual stealthiness. Additionally, we conduct human perception studies where we obtain responses from 74 anonymous participants who are engaged to evaluate whether the provided facial images that are embedded with different backdoor triggers exhibit any indications of manipulation. Each participant is presented with 5 randomly selected fake images, and 6 different triggers are applied, resulting in a total of 30 samples per participant. The ratio of identified manipulations, denoted as "IM-Ratio", for each attack method is computed based on their feedback. As shown in Table 2, our attack achieves the lowest IM-Ratio, indicating better stealthiness. Overall, our backdoor attack achieves better visual stealthiness compared to other methods in terms of qualitative, quantitative, and human perception studies, which indicates its high potential in practice.

## 5.3 ANALYSIS

**Ablations on the Kernel Sizes.** The key aspect of the proposed method is to maximize the discrepancy between the translated trigger and the original trigger, which can be quantified by convolving with a specific kernel, *i.e.*, $K(v)$. A larger kernel size implies an emphasis on maximizing the ex-

| dataset (train → test) | FF++ → FF++ | | FF++ → CDF | | FF++ → DFD | |
|---|---|---|---|---|---|---|
| kernel size | AUC | BD-AUC | AUC | BD-AUC | AUC | BD-AUC |
| $3 \times 3$ | 91.92 | 77.48 | 93.39 | 97.00 | 88.98 | 66.13 |
| $5 \times 5$ | 92.06 | **84.52** | 93.74 | 97.38 | 89.71 | **79.58** |
| $7 \times 7$ | 91.23 | 83.82 | 93.87 | 97.91 | 88.75 | 73.98 |
| $9 \times 9$ | 91.23 | 81.96 | 93.90 | **98.25** | 88.63 | 72.69 |
| $11 \times 11$ | 91.24 | 78.10 | 93.92 | 96.31 | 88.52 | 68.81 |
| $13 \times 13$ | 91.53 | 77.69 | 94.37 | 95.90 | 88.64 | 69.93 |

Table 3: Ablation study of the size of kernel $K(v)$ used to optimize the trigger generator.

pectation of the discrepancy over a broader range of translations. Here, we investigate the impact of the kernel size. We train different trigger generators using kernel sizes ranging from $3 \times 3$ to $13 \times 13$. Subsequently, we evaluate the attack performance of the triggers generated by these generators on SBI, respectively. As shown in Table 3, with the increase in kernel size, the attack performance first increases and then declines. This is probably because current detection methods typically reproduce blending artifacts by translating within a relatively small range. When the kernel size is increased, it implies the trigger is optimized over a broader translation range, which may lead to a drop in performance due to the mismatch. Therefore, we set kernel size to $5 \times 5$ in our main experiments.

**Resistance to Backdoor Defenses.** We then evaluate the resistance of our attack against backdoor defenses, *i.e.*, Fine-Tuning (FT) (Wu et al., 2022), Fine-Pruning (FP) (Liu et al., 2018), NAD (Li et al., 2021b) and ABL (Li et al., 2021a). For the backdoor defense setup, we follow the setting demonstrated in the benchmark (Wu et al., 2022). The experiments are performed on SBI, utilizing EfficientNet-b4 (Tan & Le, 2019) as the backbone network. Specifically, for FT, we fine-tune the backdoored model using $5\%$ clean data; for FP, we prune $99\%$ of the neurons in the last convolutional layer of the model and subsequently fine-tune the pruned model on $5\%$ clean data; for NAD, we use the backdoored model fine-tuned on $5\%$ clean data as the teacher model, and implement distillation on the original backdoored model; for ABL, we isolate $1\%$ of suspicious data and conduct the backdoor unlearning using the default setting.

As shown in Table 4, we can **observe**: ❶ Classical backdoor defense methods cannot provide an effective defense against our proposed attack. Even after applying defenses, the BD-AUC values still exceed $81\%$, indicating that fake faces embedded with the trigger still have a higher probability of being classified as `real`. ❷ We calculate the average prediction scores (SC) for fake faces with and without embedded triggers. A lower SC indicates a higher confidence in classification as `real`, and vice versa. The SC of fake images significantly decreases when the trigger is embedded, and even after applying backdoor defenses, it remains at a low value. This demonstrates the efficacy of our proposed method and its promising ability to evade backdoor defenses.

| dataset | FF++ → FF++ | | | |
|---|---|---|---|---|
| defense | AUC | BD-AUC | SC (w/ t) | SC (w/o t) |
| original | 92.06 | 84.52 | 15.97 | 55.75 |
| FT | 92.07 | 83.23 | 14.46 | 52.06 |
| FP | 91.74 | 85.28 | 11.96 | 51.27 |
| NAD | 92.02 | 86.05 | 15.24 | 58.72 |
| ABL | 91.07 | 81.22 | 16.74 | 53.49 |

Table 4: Evaluation of the proposed attack on backdoor defenses. "SC (w/o t)" represents the average prediction score of fake images without triggers. "SC (w/ t)" represents the score of fake images with triggers generated by our attack.

# 6 CONCLUSION

This paper introduces a novel and previously unrecognized threat in face forgery detection scenarios caused by backdoor attacks. By embedding backdoors into models and incorporating specific trigger patterns into the input, attackers can deceive detectors into producing erroneous predictions for fake images. To achieve this goal, this paper proposes *Poisoned Forgery Face* framework, a clean-label backdoor attack framework on face forgery detectors. Extensive experiments demonstrate the efficacy of our approach, and we outperform SoTA backdoor baselines by large margins. In addition, our attack exhibits promising performance against backdoor defenses. We hope our paper can draw more attention to the potential threats posed by backdoor attacks in face forgery detection scenarios.

## 7 Ethical Statement

This study aims to uncover vulnerabilities in face forgery detection caused by backdoor attacks, while adhering to ethical principles. Our purpose is to improve system security rather than engage in malicious activities. We seek to raise awareness and accelerate the development of robust defenses by identifying and highlighting existing vulnerabilities in face forgery detection. By exposing these security gaps, our goal is to contribute to the ongoing efforts to secure face forgery detection against similar attacks, making them safer for broader applications and communities.

## 8 Acknowledgement

This work is supported in part by the National Key R&D Program of China (Grant No. 2022ZD0118100), in part by National Natural Science Foundation of China (No.62025604), in part by Shenzhen Science and Technology Program (Grant No. KQTD20221101093559018).

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

# A APPENDIX

## A.1 DERIVATION OF EQUATION 8

Starting with the original optimization object for trigger generation presented in Equation 7, we have

$$
\mathbb{E}_{m,n}\big\|T_2^{m,n}(\delta) - \delta\big\|_1
$$

$$
=\frac{1}{(2v+1)^2}\sum_{m=-v}^{v}\sum_{n=-v}^{v}\big\|\delta - T_2^{m,n}(\delta)\big\|_1
$$

$$
\geq\frac{1}{(2v+1)^2}\big\|\sum_{m=-v}^{v}\sum_{n=-v}^{v}(\delta - T_2^{m,n}(\delta))\big\|_1
$$

$$
=\frac{1}{(2v+1)^2}\big\|(2v+1)^2\cdot\delta - \sum_{m=-v}^{v}\sum_{n=-v}^{v}T_2^{m,n}(\delta)\big\|_1
$$

The translated trigger $T_2^{m,n}(\delta)$ can be obtained by convolving the trigger $\delta$ with a kernel $k_{m,n}$ of shape $(2v+1)\times(2\cdot v+1)$. All values in the kernel $k_{m,n}$ are set to zero except for the element at $(v+m, v+n)$, which is set to 1, *i.e.*:

$$
k_{m,n}=
\begin{bmatrix}
0 & \cdots & \cdots & 0\\
\vdots & \ddots & & \vdots\\
\vdots & & 1 & \vdots\\
0 & \cdots & \cdots & 0
\end{bmatrix}
$$

Then we have

$$
T_2^{m,n}(\delta) = k_{m,n}\otimes\delta
$$

Specifically, the original trigger $\delta$ can be viewed as the convolution between itself and an identity kernel, *i.e.*, $\delta = k_{0,0}\otimes\delta$. Consequently, we have

$$
\frac{1}{(2v+1)^2}\big\|(2v+1)^2\cdot\delta - \sum_{m=-v}^{v}\sum_{n=-v}^{v}T_2^{m,n}(\delta)\big\|_1
$$

$$
=\frac{1}{(2v+1)^2}\big\|(2v+1)^2\cdot k_{0,0}\otimes\delta - \sum_{m=-v}^{v}\sum_{n=-v}^{v}k_{m,n}\otimes\delta\big\|_1
$$

$$
=\frac{1}{(2v+1)^2}\big\|\big[(2v+1)^2\cdot k_{0,0} - \sum_{m=-v}^{v}\sum_{n=-v}^{v}k_{m,n}\big]\otimes\delta\big\|_1
$$

$$
=\frac{1}{(2v+1)^2}\big\|K(v)\otimes\delta\big\|_1
$$

where $K(v)$ represents a convolutional kernel with a shape of $(2v+1)\times(2v+1)$, given by:

$$
\begin{bmatrix}
-1 & \cdots & -1\\
\vdots & n & \vdots\\
-1 & \cdots & -1
\end{bmatrix}
$$

where $n = (2v+1)^2 - 1$. Except for the element at $(v, v)$, all values in the kernel are set to -1. Since the coefficient $\frac{1}{(2v+1)^2}$ is irrelevant to the variable $\delta$, we can ignore the coefficient. Therefore, the original optimization objective presented in Equation 7 is equivalent to:

$$
\max_{\delta}\big\|K(v)\otimes\delta\big\|_1
$$

## A.2 Robustness of Attacks against Image Preprocessing

We conduct experiments to investigate the robustness of different attacks against various image pre-processing methods during testing. This includes image compression with a quality value of 50 (rangeing from 1 to 95), adjustments to brightness within the range of $(-0.2, +0.2)$, and adjust-ments to contrast within the range of $(-0.2, +0.2)$. We evaluate the performance on FF++ test set. The results are presented in the Table A.1. Firstly, we observe that image preprocessing algorithms could affect the performance of attacks. Secondly, these attacks are more sensitive to image com-pression compared to other image preprocessing algorithms. Thirdly, our attack still outperforms other attacks when using these image processing algorithms.

| Image Preprocessing → | | original | | Compression | | Brightness | | Contrast | |
|---|---|---|---|---|---|---|---|---|---|
| Model | Attack | AUC | BD-AUC | AUC | BD-AUC | AUC | BD-AUC | AUC | BD-AUC |
| | w/o attack | 92.32 | - | 86.80 | - | 92.02 | - | 91.90 | - |
| | Badnet | 92.47 | 48.47 | 86.69 | 49.32 | 92.04 | 48.90 | 91.88 | 48.95 |
| | Blended | 91.76 | 68.13 | 86.41 | 56.96 | 91.53 | 67.72 | 91.38 | 66.18 |
| SBI | ISSBA | 92.60 | 51.07 | 86.71 | 50.33 | 92.34 | 51.03 | 92.17 | 50.76 |
| | SIG | 91.65 | 61.18 | 85.87 | 59.78 | 91.10 | 60.53 | 91.26 | 60.73 |
| | LC | 92.17 | 61.59 | 86.43 | 60.87 | 92.08 | 60.31 | 91.87 | 61.31 |
| | **Ours** | 92.06 | **84.52** | 86.50 | **78.41** | 91.81 | **82.95** | 91.65 | **83.97** |

Table A.1: Evaluation of the robustness of backdoor attacks against different image preprocessing algorithms.

## A.3 Limitations of the Proposed Attack

In this section, we discussion the limitations of the proposed attack. Firstly, the proposed methods are not optimal. While our methods focus on the common transformations in face forgery and generally demonstrate effectiveness across different methods, they are not the optimal for specific methods. Secondly, the attack performance is partially dependent on the size of the kernel used for trigger generation. The choice of kernel size can have an impact on the effectiveness of the attack.

## A.4 Implementation of the Generator

We provide the detailed network architecture of the generator $G(z)$, in Figure A.1. The generator is trained using the loss function $L_g$, as presented in Equation 10.

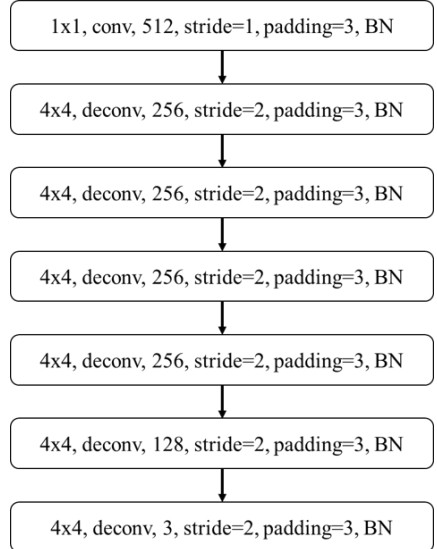

Figure A.1: The network architecture of the generator.

## A.5 ABLATION ON TRIGGER EMBEDDING

In this section, we conduct an ablation study to investigate the impact of the hyperparameter $\alpha$. This involves varying the blending type (relative or absolute) and the blending ratio (value of $a$). The results are presented in the Table A.2. In the table, the 'relative' line represents the method we used in our paper, where $\alpha$ is defined as $\alpha = a \cdot x_k/255$. On the other hand, the 'absolute' line indicates the experiment we conducted as a reference, where $\alpha$ is defined as $\alpha = a$. From the results, we can draw observations as follows: 1) As the value of $a$ increases, we can achieve higher attack performance but lower PSNR. This is because the tradeoff between the attack performance and the stealthiness of the backdoor trigger. 2) Although absolute embedding yields better attack performance, this improvement comes at the cost of lower PSNR.

| type | $a$ | AUC | BD-AUC | PSNR |
|---|---|---|---|---|
| relative | 0.03 | 91.78 | 67.02 | 39.56 |
| | 0.05 | 92.06 | 84.52 | 35.19 |
| | 0.07 | 91.63 | 86.81 | 32.30 |
| | 0.1 | 91.31 | 92.61 | 29.22 |
| absolute | 0.05 | 91.50 | 92.16 | 29.71 |

Table A.2: Ablation on the embedding type and embedding ratio.

## A.6 ADDITIONAL COMPARISONS WITH FREQUENCY-BASED BACKDOOR ATTACKS

In this section, we extend our comparisons to include a frequency-based baseline, FTrojan (Wang et al., 2022a). The results are presented in Table A.3. Our method outperforms FTrojan across all three detectors and three datasets.

| Dataset (train → test) | | | FF++ → FF++ | | FF++ → CDF | | FF++ → DFD | |
|---|---|---|---|---|---|---|---|---|
| Type | Model | Attack | AUC | BD-AUC | AUC | BD-AUC | AUC | BD-AUC |
| Deepfake artifact detection | Xception | w/o attack | 85.10 | - | 77.84 | - | 76.85 | - |
| | | FTrojan | 84.56 | 94.51 | 76.19 | 96.25 | 78.04 | 92.80 |
| | | **Ours** | 85.18 | **99.65** | 77.21 | **99.13** | 78.26 | **95.89** |
| Blending artifact detection | SBI | w/o attack | 92.32 | - | 93.10 | - | 90.35 | - |
| | | FTrojan | 92.40 | 65.54 | 93.61 | 87.38 | 89.97 | 59.45 |
| | | **Ours** | 92.06 | **84.52** | 93.74 | **97.38** | 89.71 | **79.58** |
| | Face X-ray | w/o attack | 78.90 | - | 85.38 | - | 83.30 | - |
| | | FTrojan | 78.02 | 47.26 | 82.04 | 54.78 | 82.21 | 56.03 |
| | | **Ours** | 77.70 | **79.82** | 81.74 | **98.96** | 83.52 | **98.55** |

Table A.3: Comparisons with frequency-based backdoor attack, FTrojan.

