# OpenReview forum: "Poisoned Forgery Face: Towards Backdoor Attacks on Face Forgery Detection"
_ICLR.cc/2024/Conference — ICLR 2024 spotlight_

### Official Review · Reviewer_GcXY · 2023-10-28

**Soundness:** 3 good
**Presentation:** 3 good
**Contribution:** 3 good
**Rating:** 6
**Confidence:** 4

**Summary:**

This paper aims to attack face deepfake detection models via backdoor attack, especially for blending artifact detection methods. To resolve the scalable problem, the paper builds a trigger generator to generate the trigger adaptively. To ensure the stealthiness of the trigger, a relative pixel-wise embedding ratio is incorporated in the generation of the poisoned sample. In general, the style of this paper is clear and logical. However, some experiments are missing and hence make readers confused about the effectiveness of some modules.

**Strengths:**

1. The paper is the early exploration of backdoor attacks in face forgery detection and the description of the existing obstacles deployed trigger in face forgery detection is clear.
2. Besides, the paper subtly simplifies the complex forgery problem as linear translation transformation and hence makes the backdoor attack in the translation-sensitive scenario.
3. To make the trigger stealthy, the author further hid the generated noise by a simple weight parameter.

**Weaknesses:**

1.Some descriptions like formulation may be incorrect and may misunderstand the readers.
2.Some ablation studies are missing and make readers confused about the effectiveness of the proposed modules.
3.The method lacks generalization to some extent. The main gains derive from blending artifact detection. However, in the real scenario, not all evidence of determining forgery is based on blending traces in face forgery detection. Therefore, I would like to see the feedback about the questions listed below and decide to give a reasonable rating.

**Questions:**

The current rating is not my final decision, I hope the authors can explan the questions listed below in detail:

1.In Eq. (2), as stated, the blending transformation needs two real samples, however, the variable in function T^b only has one sample instead of a pair of real samples. For example, in Face X-ray, the fake face image is generated via two different real ones. So I think the formulation should be adjusted to make it more reasonable.

2.In the section on ‘Stealthiness of Backdoor Attacks’, Table 2 shows the qualitative results among different backdoor attack methods. However, the paper does not mention the number of samples for evaluation.

3.The influence of the hyperparameter ‘a’ is ambiguous. Moreover, the comparison with/without ‘alpha’ is missing and hence makes readers confused about the effectiveness brought by the proposed relative embedding. It is better to show the qualitative results and the quantitative backdoor attack performance.

4.This attack method is built on the generated image should show a relatively distinct facial boundary. However, in real scenarios, fake images are generated from various transformations. If some fake images are generated only via non-linear transformation, such as blur, this method may not be optimal. This may be the reason why the attack performance is lower than ‘LC’ because not all training samples contain linear translation transformation.

5.Typo: In Resistance to Backdoor Defenses, ‘Efficient-b4’ should be ‘EfficientNet-b4’.

---

> ### Author Response · Authors · 2023-11-15
> **Official Response to Reviewer GcXY (1/2)**
>
> The author would like to thank the reviewer for appreciating our clear problem formulation and the effectiveness of our proposed attack. We provide more evaluation detail, conduct additional ablation, and explain the motivation behind our method, which we hope can address your concerns.
>
> **Q1**: In Eq. (2), as stated, the blending transformation needs two real samples, however, the variable in function T^b only has one sample instead of a pair of real samples. For example, in Face X-ray, the fake face image is generated via two different real ones. So I think the formulation should be adjusted to make it more reasonable.
>
> **A1**: Thanks for your suggestions. Our current formulation is based on the SBI method where the two real samples are from the same source, which can be considered as a special case. We appreciate your feedback and we have revised the formulation to a more general form that can represent different blending artifact methods, including SBI and Face X-ray, etc.
>
> **Q2**: In the section on ‘Stealthiness of Backdoor Attacks’, Table 2 shows the qualitative results among different backdoor attack methods. However, the paper does not mention the number of samples for evaluation.
>
> **A2**: Thanks for your feedback. In the 'Stealthiness of Backdoor Attacks' section, we conducted a quantitative (PSNR and L_inf) evaluation using the fake subset of the FF++ dataset's test set. This subset consists of 560 fake videos, and we extracted 32 frames per video, resulting in a total of 17,920 samples. For the human perception studies, we randomly selected 5 images from the aforementioned subset and applied all 6 attacks, resulting in a total of 30 samples per participant. We have included these details in the revision.
>
> **Q3**: The influence of the hyperparameter ‘a’ is ambiguous. Moreover, the comparison with/without ‘alpha’ is missing and hence makes readers confused about the effectiveness brought by the proposed relative embedding. It is better to show the qualitative results and the quantitative backdoor attack performance.
>
> **A3**: Thanks for your feedback. We would like to address your concerns as follows:
> 1) The hyperparameter 'a' serves as a scaler to to determine the maximum magnitude of the backdoor trigger. A higher 'a' value results in a less stealthy trigger, while a lower 'a' value enhances stealthiness. In our experiment, we set 'a' to a low value (0.05) to prioritize stealthiness of the trigger.
> 2) The hyperparameter 'alpha' is defined as '**alpha**' = a * **x_k** / 255, where 'x_k' represents the pixel values of the image. When embedding the trigger into the image, we adjust the pixel values of the trigger based on the magnitude of the pixel values in the image. This adjustment further enhances the stealthiness of the trigger.
> 3) We clarify that training without 'alpha' or setting 'alpha' to zero is equivalent to benign training without any backdoor attack. In our paper, we have presented the results of benign training in the 'w/o attack' line of Table 1. Furthermore, we are happy to follow your suggestions and conduct an extra ablation study to investigate the impact of the hyperparameter 'alpha'. This involves varying the blending type (relative or absolute) and the blending ratio (value of 'a') in the table mentioned. In the table, the 'relative' line represents the method we used in our paper, where 'alpha' is defined as '**alpha**' = a * **x_k** / 255. On the other hand, the 'absolute' line indicates the experiment we conducted as a reference, where 'alpha' is defined as '**alpha**' = a.
> 4) From the results, we can draw observations as follows:
>     1)  As the value of 'a' increases, we can achieve higher attack performance but lower PSNR. This is because **the tradeoff between the attack performance and the stealthiness of the backdoor trigger**.
>     2)  Although absolute embedding yields better attack performance, this improvement comes at the cost of lower PSNR.
>
> |   type   | 'a'  | AUC   | BD-AUC | PSNR  |
> |:--------:|------|-------|--------|-------|
> | relative | 0.03 | 91.78 | 67.02  | 39.56 |
> | relative | 0.05 | 92.06 | 84.52  | 35.19 |
> | relative | 0.07 | 91.63 | 86.81  | 32.30 |
> | relative | 0.1  | 91.31 | 92.61  | 29.22 |
> | absolute | 0.05 | 91.50 | 92.16  | 29.71 |

---

> > ### Author Response · Authors · 2023-11-15
> > **Official Response to Reviewer GcXY (2/2)**
> >
> > **Q4**: This attack method is built on the generated image should show a relatively distinct facial boundary. However, in real scenarios, fake images are generated from various transformations. If some fake images are generated only via non-linear transformation, such as blur, this method may not be optimal. This may be the reason why the attack performance is lower than ‘LC’ because not all training samples contain linear translation transformation.
> >
> > **A4**: Thanks for your feedback. We acknowledge that our methods are not optimal in all cases and we would like to offer the following insights:
> >
> > 1) In the case of blending artifact methods, where fake images are created by blending two real faces during training, our attack is built upon the idea that the effectiveness of a backdoor attack relies on maximizing the discrepancy between the trigger in the real face and the transformed trigger in the fake face. The generation of fake faces involves various transformations, and different methods employ different compositions of these transformations. In our approach, we target a transformation that causes distinct facial boundaries, as it is prevalent across different blending artifact methods. Thereby, our attack demonstrates transferability across different methods. In the case where fake images are generated only through non-linear transformations, such as blur, our method may not be optimal. However, the underlying idea can be extended in such scenarios. An ideal approach would be to select the representative non-linear transformation as the target and optimize a trigger that maximizes the difference between the original version and the transformed version of the trigger. In addition, a more general trigger optimization that can adapt to more face forgery detection methods is a future research direction and we will take into consideration.
> > 2) The potential reason for the lower performance compared to 'LC' in Xception could be attributed to the higher stealthiness achieved by our method at the cost of sacrificing attack performance. The focus on enhancing stealthiness may result in a trade-off with the overall attack performance. Here, we conduct additional experiments where we enhance the trigger embedding ratio, which refers to the magnitude of the trigger. From the results in the table below, as we increase the embedding ratio, our method outperforms LC while remaining PSNR higher than LC.
> >
> > |        |                 |  FF++ |  FF++  |  CDF  |   CDF  |  DFD  |   DFD  |       |
> > |:------:|:---------------:|:-----:|:------:|:-----:|:------:|:-----:|:------:|:-----:|
> > | attack | embedding ratio |  AUC  | BD-AUC |  AUC  | BD-AUC |  AUC  | BD-AUC |  PSNR |
> > |   LC   |        -        | 84.25 |  99.97 | 75.29 |  99.58 | 77.99 |  99.36 | 20.51 |
> > |  Ours  |       0.05      | 85.18 |  99.65 | 77.21 |  99.13 | 78.26 |  95.89 | 35.19 |
> > |  Ours  |       0.15      | 85.16 |  99.98 | 76.72 |  99.81 | 77.27 |  99.39 | 26.86 |
> >
> > **Q5**: Typo: In Resistance to Backdoor Defenses, ‘Efficient-b4’ should be ‘EfficientNet-b4’.
> >
> > **A5**: Thanks for your correction. We have revised in the revision.

---

> ### Comment · Reviewer_GcXY · 2023-11-23
> **Response to the authors**
>
> I wish to convey my appreciation for your considerate responses and meticulous revisions. However, some of the papers related to face forgery detection should be discussed in the related work as previous detection methods is relatively old:
>
> [1] Shao R, Wu T, Liu Z. Detecting and recovering sequential deepfake manipulation[C]//European Conference on Computer Vision. Cham: Springer Nature Switzerland, 2022: 712-728.
>
> [2] Xia R, Liu D, Li J, et al. MMNet: Multi-Collaboration and Multi-Supervision Network for Sequential Deepfake Detection[J]. arXiv preprint arXiv:2307.02733, 2023.
>
> [3] Shao R, Wu T, Liu Z. Robust Sequential DeepFake Detection[J]. arXiv preprint arXiv:2309.14991, 2023.

---

> > ### Author Response · Authors · 2023-11-23
> > **Official Comments by Authors**
> >
> > Dear Reviewer GcXY,
> >
> > We really appreciate your constructive comments and suggestions. The paper's quality has been greatly enhanced by your tremendous efforts. We followed your suggestions and discussed/cited these forgery face detection methods as the related work in Section 2 in our latest revision.
> >
> > Best regards,
> >
> > Authors of submission 2182

---

### Official Review · Reviewer_4qiC · 2023-10-30

**Soundness:** 4 excellent
**Presentation:** 4 excellent
**Contribution:** 4 excellent
**Rating:** 8
**Confidence:** 5

**Summary:**

In this paper, a novel backdoor attack strategy is introduced for the FACE FORGERY DETECTION task. This approach deceives face forgery detectors by embedding specific triggers within the training data. The authors delve deeply into the internal structure of current face forgery detectors and propose a transformation-sensitive triggering mechanism to facilitate effective backdoor attacks. Extensive experiments validate the efficacy of the proposed method and highlight potential vulnerabilities in face forgery detection systems.

**Strengths:**

1. The paper is the first to uncover the vulnerabilities during the training phase of face forgery detection, introducing a practical clean-label attack technique.

2. The backdoor attack strategy presented is notably innovative, leveraging the unique architectural characteristics of face forgery methods.

3. The trigger designed in this study is highly covert, making it challenging for the human eye to detect.

4. The authors provide a comprehensive explanation of the motivations behind the proposed method, and the manuscript is articulated clearly.

**Weaknesses:**

1. The paper could provide a discussion on the limitations of the proposed method, ideally placed in the appendix.
2. The choice of different kernel sizes for the trigger can impact the attack performance on various forgery detection models. The rationale behind such an impact remains unclear. Additionally, the authors' decision to finalize a size of 5 for the kernel lacks a clear justification.
3. Table 4 indicates that existing defense mechanisms fail against the proposed attack. It remains ambiguous whether this failure is specific to the attack introduced by the authors or if it's a general shortcoming for other attacks as well. Furthermore, it would be valuable if the authors could analyze existing attacks to suggest potential defense strategies.
4. In the section on trigger generation, the authors should provide a comprehensive description of the generator's overall loss function and its basic architecture.

**Questions:**

Please refer to [Weakness].

---

> ### Author Response · Authors · 2023-11-15
> **Official Response to Reviewer 4qiC (1/2)**
>
> The author would like to thank the reviewer for appreciating our motivation and high steathiness and performance of our proposed attack. We provide detailed explanation of mechanism behind kernel size and further evaluate existing defense and discuss potential defense strategies, which we hope can address your concerns.
>
> **Q1**: The paper could provide a discussion on the limitations of the proposed method, ideally placed in the appendix.
>
> **A1**: Thanks for your suggestions. We would like to discuss the limitations of the proposed method as follows:
>
> 1) The proposed methods are not optimal. While our methods focus on the common transformations in face forgery and generally demonstrate effectiveness across different methods, they are not the optimal for specific methods.
> 2) The attack performance is partially dependent on the size of the kernel used for trigger generation. The choice of kernel size can have an impact on the effectiveness of the attack.
>
> Finally, we have included this discussion to the appendix.
>
> **Q2**: The choice of different kernel sizes for the trigger can impact the attack performance on various forgery detection models. The rationale behind such an impact remains unclear. Additionally, the authors' decision to finalize a size of 5 for the kernel lacks a clear justification.
>
> **A2**: Thanks for your suggestions. We would address your concern as follows:
> 1) As mentioned in Section 5.3 of the paper, the choice of kernel size indicates the range of translation that we aim to optimize. This range is based on the attacker's assumption on the translation amplitude during the synthesis of fake data. A larger kernel size suggests that the triggers are optimized over a broader translation range. However, a larger kernel size may not always lead to better performance. This is because the triggers could be optimized to adapt to a larger range of translations, which may not be the best match for the translation amplitude during the synthesis of fake data.
> 2) Regarding the decision to use a relatively small kernel size of 5, this is based on the consideration that the distinction in facial boundaries of the synthesized fake images can be really small, in order to make the synthesized fake images appear more natural and less suspicious to the human eye.
>
> **Q3**: Table 4 indicates that existing defense mechanisms fail against the proposed attack. It remains ambiguous whether this failure is specific to the attack introduced by the authors or if it's a general shortcoming for other attacks as well. Furthermore, it would be valuable if the authors could analyze existing attacks to suggest potential defense strategies.
>
> **A3**: Thanks for your suggestions. Firstly, we conduct additional experiments to evaluate the performance of existing defense methods against the current attack of best performance, specifically the blended attack. These experiments are conducted on the **SBI** detection method on FF++ dataset, and the results are presented in the following table. From the results, we observe that under the clean label setting, **the four defense methods tested also fail to effectively defend against the existing attack** in face forgery detection. This finding suggests that while attacking blending artifact methods is more challenging, once a successful attack is launched, it establishes a strong backdoor connection that existing defense mechanisms may struggle to mitigate.
>
> |  defense |  AUC  | BD-AUC | SC (w/ t) | SC (w/o t) |
> |:--------:|:-----:|:------:|:---------:|:----------:|
> | original | 91.76 |  68.13 |   33.10   |    54.06   |
> |    FT    | 91.11 |  68.01 |   30.06   |    57.88   |
> |    FP    | 91.74 |  69.98 |   20.46   |    49.63   |
> |    NAD   | 91.43 |  67.96 |   27.74   |    57.48   |
> |    ABL   | 91.55 |  66.42 |   35.72   |    56.62   |
>
> Regarding the **potential defense strategies**, we would like to offer the following insights:
> 1) During our fine-pruning defense experiments, we observed that the neural activation in the linear layer is sparse. For example, we selectively prune a significant portion (e.g., 99%) of the less-contributing neurons and fine-tune the model on clean data. Even with such aggressive pruning, the model is able to maintain performance. This result inspires a potential future direction in which we could explore more fine-grained pruning techniques on the remaining 1% of neurons.
> 2) In the clean label setting, where only the images are modified while the labels remain unchanged, we hypothesize that adding noise to the images during training can help destroy the trigger pattern and mitigate the establishment of the backdoor connection. This approach may be particularly effective when the magnitude of the backdoor trigger is kept small in order to ensure stealthiness.

---

> ### Author Response · Authors · 2023-11-15
> **Official Response to Reviewer 4qiC (2/2)**
>
> **Q4**: In the section on trigger generation, the authors should provide a comprehensive description of the generator's overall loss function and its basic architecture.
>
> **A4**: Thank you for your suggestion. We would like to provide a more detailed description of the generator's basic architecture and the overall loss function:
>
> 1) **Basic Architecture** of the Generator (G):
>
> - 1x1, conv, 512, stride=1, padding=3, BN
> - 4x4, deconv, 256, stride=2, padding=3, BN
> - 4x4, deconv, 256, stride=2, padding=3, BN
> - 4x4, deconv, 256, stride=2, padding=3, BN
> - 4x4, deconv, 256, stride=2, padding=3, BN
> - 4x4, deconv, 128, stride=2, padding=3, BN
> - 4x4, deconv, 3, stride=2, padding=3, BN
>
> In the above description, "KxK, conv/deconv, C, stride=S, padding=P, BN" indicates a convolutional or deconvolutional layer with a KxK kernel, C output filters, stride S, and padding P, followed by batch normalization (BN).
>
> 2) **Overall Loss Function** for the Generator: The overall loss function for the generator G(z) can be found in Equation 10 of the paper. It is represented as L_g = − log ||K(v)⊗G(z)||_1. Here, K(v) denotes our proposed convolutional kernel as described in Equation 8, and G(z) represents the output of the generator with a latent variable z ~ N(0,1) as input. We train the generator G(z) with L_g as the loss function.

---

> > ### Comment · Reviewer_4qiC · 2023-11-21
> > **Official Comment by Reviewer 4qiC**
> >
> > I appreciate the authors' feedbacks, extra experiments and explanations. Based on the response, I would like to keep my recommendation for acceptance.

---

> > > ### Author Response · Authors · 2023-11-22
> > > **Official Comments by Authors**
> > >
> > > Dear Reviewer 4qiC,
> > >
> > > We sincerely thank you for your valuable comments and positive support. The paper's quality has been greatly enhanced by your tremendous efforts.
> > >
> > > Appreciated!
> > >
> > > Best regards,
> > >
> > > Authors of submission 2182

---

### Official Review · Reviewer_uGDS · 2023-11-01

**Soundness:** 3 good
**Presentation:** 3 good
**Contribution:** 3 good
**Rating:** 8
**Confidence:** 4

**Summary:**

The paper proposes the Poisoned Forgery Face clean label backdoor attack framework utilizing a mask for the inner face region and a stealthy translation-sensitive trigger pattern, which is suitable for attacking facial forgery detectors. By poisoning a small portion of the clean training data, the attacker can flip the label of deep fake images or videos from fake to real. This approach can overcome the label conflict problem, which happens when the detectors are trained with self-blending techniques with only real data. Experimental results demonstrated a significant improvement in the attack success rate and the improvement of the trigger stealthiness.

**Strengths:**

+ The paper addresses not only the traditional deepfake artifact detection, in which detectors are trained with deepfakes, but also the recent blending artifact detection, in which the detectors are trained with augmented self-blended real images. The proposed translation-sensitive trigger pattern is innovative and effective for dealing with the self-blending approach.

+ The paper also provides a comprehensive benchmark for backdoor attacks, which is not available in the literature.

**Weaknesses:**

+ This paper is not the first in the literature to address backdoor attacks in face forgery detection. Cao et al. [A] have already investigated this problem. Therefore, the first contribution is invalid.

+ The use of "b" in the equations in section 3 is confusing. b presents both "blending" and "remaining clean."

+ The benchmark should include some frequency-based backdoor attacks like [B].

+ The reported performance of Face X-ray is lower than that in the original paper and in the SBI paper. I am wondering if there is something wrong with the experiments.

+ The robustness of the proposed trigger patterns was not investigated. While sharing deepfake images or videos on social networks, these patterns may be destroyed by compression or some other image processing algorithms.

+ The paper should include a paragraph of ethics statement

+ The appendix section, which was referred from the main part, is missing.

References:

[A] Cao, Xiaoyu, and Neil Zhenqiang Gong. "Understanding the security of deepfake detection." In International Conference on Digital Forensics and Cyber Crime, pp. 360-378. Cham: Springer International Publishing, 2021.

[B] Wang, Tong, Yuan Yao, Feng Xu, Shengwei An, Hanghang Tong, and Ting Wang. "An invisible black-box backdoor attack through frequency domain." In European Conference on Computer Vision, pp. 396-413. Cham: Springer Nature Switzerland, 2022.

**Questions:**

Please refer to the comments in the weaknesses section.

---

> ### Author Response · Authors · 2023-11-15
> **Official Response to Reviewer uGDS (1/3)**
>
> The author would like to thank the reviewer for appreciating our comprehensive benchmark and our proposed attack that is effective across different face forgery methods. We add additional benchmark on frequency-based attack and evaluate the robustness against various image preprocessing algorithms, which we hope can address your concerns.
>
> **Q1**: This paper is not the first in the literature to address backdoor attacks in face forgery detection. Cao et al. [A] have already investigated this problem. Therefore, the first contribution is invalid.
>
> **A1**: Thank you for your correction. We have thoroughly reviewed the mentioned paper by Cao et al. [A] and acknowledge its contribution to backdoor attacks in face forgery detection. We apologize for any confusion caused by our previous statement regarding the "first" contribution. Additionally, we would like to highlight the distinctions and emphasis of our paper in comparison to the mentioned work:
> 1) **Scope of investigation**: While the mentioned paper focuses on investigating a single backdoor attack, specifically Badnet, in face forgery detection, our paper goes beyond and provides a more comprehensive analysis. We benchmark five different backdoor attacks and propose a novel attack method in this field.
> 2) **Extension to blending artifact detection methods**: Our research also extends to the examination of backdoor attacks on blending artifact detection methods, specifically SBI and Face X-ray. These methods are more challenging to attack compared to the previously studied detection methods.
>
> Finally, we have made the necessary modifications and added appropriate citations in the revision.
>
> **Q2**: The use of "b" in the equations in section 3 is confusing. b presents both "blending" and "remaining clean."
>
> **A2**: Thank you for pointing out the issue. In the revision, we clarify and differentiate the symbols used to represent "blending" and "remaining clean". Specifically, we use "b" to represent "blending" and introduce 'c' to represent "remaining clean".
>
> **Q3**: The benchmark should include some frequency-based backdoor attacks like [B].
>
> **A3**: We appreciate your suggestion and conduct experiments that involve the mentioned frequency-based attack, FTrojan [B]. The results are presented in the following table. Upon analysis, our attack outperforms the frequency-based attack, FTrojan [B], on three face forgery detection models.
>
> | **Dataset** |    **→**   |   FF++  |    FF++    |   CDF   |     CDF    |   DFD   |     DFD    |
> |:-----------:|:----------:|:-------:|:----------:|:-------:|:----------:|:-------:|:----------:|
> |  **Model**  | **Attack** | **AUC** | **BD-AUC** | **AUC** | **BD-AUC** | **AUC** | **BD-AUC** |
> |   Xception  |   FTrojan  |  84.56  |    94.51   |  76.19  |    96.25   |  78.04  |    92.80   |
> |   Xception  |  **Ours**  |  85.18  |  **99.65** |  77.21  |  **99.13** |  78.26  |  **95.89** |
> |     SBI     |   FTrojan  |  92.40  |    65.54   |  93.61  |    87.38   |  89.97  |    59.45   |
> |     SBI     |  **Ours**  |  92.06  |  **84.52** |  93.74  |  **97.38** |  89.71  |  **79.58** |
> |  Face X-ray |   FTrojan  |  78.02  |   47.26    |  82.04  |    54.78   |  82.21  |    56.03   |
> |  Face X-ray |  **Ours**  |  77.70  |  **79.82** |  81.74  |  **98.96** |  83.52  |  **98.55** |

---

> > ### Author Response · Authors · 2023-11-15
> > **Official Response to Reviewer uGDS (3/3)**
> >
> > **Q5**: The robustness of the proposed trigger patterns was not investigated. While sharing deepfake images or videos on social networks, these patterns may be destroyed by compression or some other image processing algorithms.
> >
> > **A5**: Thank you for your suggestion. We conduct additional experiments to investigate the robustness of different attacks against various image preprocessing methods during testing. This includes image compression, adjustments to brightness, and adjustments to contrast. We evaluate the performance on SBI on the FF++ test set. The results are presented in the following table.
> > 1) We observe that image preprocessing algorithms could affect the performance of attacks.
> > 2) These attacks are more sensitive to image compression compared to other image preprocessing algorithms.
> > 3) Our attack still outperforms other attacks when using these image processing algorithms.
> >
> > |            | original |  original  | Compression | Compression | Brightness | Brightness | Contrast |  Contrast  |
> > |:----------:|:--------:|:----------:|:-----------:|:-----------:|:----------:|:----------:|:--------:|:----------:|
> > | **Attack** | **AUC**  | **BD-AUC** | **AUC**     | **BD-AUC**  | **AUC**    | **BD-AUC** | **AUC**  | **BD-AUC** |
> > |      -     |   92.32  |      -     |    86.80    |      -      |    92.02   |      -     |   91.90  |      -     |
> > |   Badnet   |   92.47  |    48.47   |    86.69    |    49.32    |    92.04   |    48.90   |   91.88  |    48.95   |
> > |   Blended  |   91.76  |    68.13   |    86.41    |    56.96    |    91.53   |    67.72   |   91.38  |    66.18   |
> > |    ISSBA   |   92.60  |    51.07   |    86.71    |    50.33    |    92.34   |    51.03   |   92.17  |    50.76   |
> > |     SIG    |   91.65  |    61.18   |    85.87    |    59.78    |    91.10   |    60.53   |   91.26  |    60.73   |
> > |     LC     |   92.17  |    61.59   |    86.43    |    60.87    |    92.08   |    60.31   |   91.87  |    61.31   |
> > |    Ours    |   92.06  |  **84.52** |    86.50    |  **78.41**  |    91.81   |  **82.95** |   91.65  |  **83.97** |
> >
> > **Q6**: The paper should include a paragraph of ethics statement
> >
> > **A6**: Thank you for your suggestions. We have incorporated the following ethics statement paragraph in the revision.
> >
> > **Ethics Statement**:
> > This study aims to uncover vulnerabilities in face forgery detection caused by backdoor attacks, while adhering to ethical principles. Our purpose is to improve system security rather than engage in malicious activities. We seek to raise awareness and accelerate the development of robust defenses by identifying and highlighting existing vulnerabilities in face forgery detection. By exposing these security gaps, our goal is to contribute to the ongoing efforts to secure face forgery detection against similar attacks, making them safer for broader applications and communities.
> >
> > **Q7**: The appendix section, which was referred from the main part, is missing.
> >
> > **A7**: We apologize for any confusion. The appendix section can be found in the attached **supplemental materials (.pdf)** in the openreview website or **in our revision**.
> >
> >
> > *References*:
> >
> > [1] Shiohara K, Yamasaki T. Detecting deepfakes with self-blended images[C]//Proceedings of the IEEE/CVF Conference on Computer Vision and Pattern Recognition. 2022: 18720-18729.
> >
> > [2] Rossler A, Cozzolino D, Verdoliva L, et al. Faceforensics++: Learning to detect manipulated facial images[C]//Proceedings of the IEEE/CVF international conference on computer vision. 2019: 1-11.

---

> > > ### Comment · Reviewer_uGDS · 2023-11-21
> > >
> > > I would like to express my gratitude for your thoughtful answers and diligent revisions. They have improved the quality of the paper. I also wanted to inform you that I have made an adjustment to the scores.

---

> > > > ### Author Response · Authors · 2023-11-21
> > > > **Official Comments by Authors**
> > > >
> > > > Dear Reviewer uGDS,
> > > >
> > > > We really appreciate your prompt feedback and the amount of time you have spent reviewing our paper! We sincerely thank you for your valuable comments and suggestions. The paper's quality has been greatly enhanced by your tremendous efforts.
> > > >
> > > > Appreciated!
> > > >
> > > > Best regards,
> > > >
> > > > Authors of submission 2182

---

> ### Author Response · Authors · 2023-11-15
> **Official Response to Reviewer uGDS (2/3)**
>
> **Q4**: The reported performance of Face X-ray is lower than that in the original paper and in the SBI paper. I am wondering if there is something wrong with the experiments.
>
> **A4**: Thank you for bringing these points to our attention. The performance difference on clean samples can be attributed to two main factors:
> 1) **Data version**: The FF++ datasets have multiple versions, including the raw (original) version and compressed versions such as c23(HQ) and c40(LQ). As mentioned in Section 4.4 of the SBI paper [1], the original performance of both Face X-ray and SBI models was evaluated on the raw version of the data. But in our experiments, we used the compressed (c23) version of the data for both training and testing. It is known that the performance on clean compressed data may experience a drop [2]. On the other hand, the CDF dataset has only one version, and our reported performance on SBI is 93.10, which is close to the results reported in the original paper (93.18) and the official GitHub repository (92.87%).
> 2) **Unified face extraction process**: To avoid possible bias in the data preprocessing, we employed a unified face extraction process proposed by SBI, which includes consistent parameters such as the number of frames per video and the area of the extracted faces, for all three face forgery detection methods. This standardized approach may lead to performance differences compared to original paper.
>
> In addition, we conducted evaluations on the **raw version** of FF++ dataset. The results are presented in the following table.
> 1) We evaluate the clean performance on the raw data, which aligns with the evaluation approach used in the original papers. The 'original w/o attack' line in the table represents the results reported in the original paper. The results indicate that our performance on clean raw data is close to the original reported results.
> 2) We also evaluate the attack performance on the raw data. We compare the existing attack with the best performance with our method on the raw data. We observe that the attack performance, as indicated by BD-AUC, is higher on the raw data compared to the compressed (c23) data. According to these results, **our method still demonstrates superior performance compared to the existing attack on the raw data**.
>
> |    Model   |        Attack       | Version |  AUC  |   BD-AUC  |
> |:----------:|:-------------------:|:-------:|:-----:|:---------:|
> |     SBI    | original w/o attack |   raw   | 99.64 |     -     |
> |     SBI    |       Blended       |   raw   | 99.38 |   78.34   |
> |     SBI    |         Ours        |   raw   | 99.46 | **97.76** |
> |     SBI    |       Blended       |   c23   | 91.76 |   68.13   |
> |     SBI    |         Ours        |   c23   | 92.06 | **84.52** |
> | Face X-ray | original w/o attack |   raw   | 98.52 |     -     |
> | Face X-ray |       Blended       |   raw   | 93.85 |   84.57   |
> | Face X-ray |         Ours        |   raw   | 94.03 | **94.42** |
> | Face X-ray |       Blended       |   c23   | 75.02 |   72.10   |
> | Face X-ray |         Ours        |   c23   | 77.70 | **79.82** |

---

### Author Response · Authors · 2023-11-15
**General Response: Revision Summary**

We would like to thank all reviewers for their valuable and constructive suggestions, which have greatly contributed to the improvement of our paper. In response, we have conducted additional experiments and included illustrations to address the raised concerns. We provide a summary of these revisions as follows:
- We revised the statement regarding “the first work” and emphasized our contribution in our revision (paragraph 2 and **contribution** in section 1).
- We revised the superscript that refers to "remaining clean" from "b" to "c" in section 3.
- We added a benchmark on a frequency-based backdoor attack, FTrojan. The results are presented in Table 1.
- We included the evaluation on the robustness of different backdoor attacks against image preprocessing algorithms in section A.2 in the Appendix.
- We included a paragraph of ethics statement in section 7.
- We included a discussion on the limitations of our methods in section A.3 in the Appendix.
- We provided a description of the network architecture of the generator in section A.4 in the Appendix.
- We revised the general formulation of the blending transformation to provide an accurate description of the blending of two faces that can be different, in Equation 2,4,5,6 in section 3,4.
- We provided details regarding the number of samples used for the quantitative evaluation of the stealthiness of the trigger in section 5.2.
- We included the ablation study on the hyperparameter for trigger embedding, 'alpha', in section A.5 in the Appendix
- We revised the typo ‘Efficient-b4’ to ‘EfficientNet-b4’ in section 5.3

All revisions in the PDF file have been highlighted in blue.

---

### Author Response · Authors · 2023-11-18
**Invitation to Reviewer-Author Discussion**

Dear reviewers,

Once again, we extend our sincere gratitude for your insightful comments and valuable advice on our paper. Your constructive feedback is vital to the enhancement of our work.

We would like to invite you for further discussion of our paper and are happy to respond to any further questions. Your active participation in the ongoing discussion will be crucial to ensuring well-informed decisions, and we deeply appreciate your consideration in dedicating further time to our paper.

Best regards,

Authors of submission 2182

---

### Comment · Area_Chair_Lsfz · 2023-11-20
**Comments on authors' responses**

Dear Reviewers,
The authors have responded to your valuable comments.
Please take a look at their responses!

---

### Author Response · Authors · 2023-11-21
**A gentle reminder: period ending– we anticipate your feedback!**

Dear reviewers,

Thank you again for your services and every minute of your time spent reviewing this paper.

As the discussion stage is closing, we sincerely look forward to your feedback. The authors deeply appreciate your valuable time and efforts spent reviewing this paper and helping us improve it. The manuscript's quality has been enhanced by the reviewers' tremendous efforts and insightful comments.

Please also let us know if there are further improvements or comments about this paper. We strive to improve the paper consistently, and it is our pleasure to have your feedback!

Best Regards,

Authors of submission 2182

---

### Meta-Review · Program_Chairs · 2023-12-05

**Metareview:**

Backdoor attack has been early proposed for image classifier model and then extended to other problems/models. This paper presents a new study of backdoor attack on face forgery detection.
All reviewers consistently appreciate the contributions of this paper and authors' responses satisfy the reviewers.
In responding to Q5 of Reviewer uGDS, it is suggested to clearly describe the parameters of each attack (e.g., quality factor of compression, contrast, etc).

**Justification For Why Not Higher Score:**

Narrow scope for an ICLR audience.

**Justification For Why Not Lower Score:**

All reviewers consistently appreciate the contributions of this paper and authors' responses satisfy the reviewers.

---

### Decision · Program_Chairs · 2024-01-16

Accept (spotlight)